# Theory and Practice of Coarse-Grained Molecular Dynamics of Biologically Important Systems

**DOI:** 10.3390/biom11091347

**Published:** 2021-09-11

**Authors:** Adam Liwo, Cezary Czaplewski, Adam K. Sieradzan, Agnieszka G. Lipska, Sergey A. Samsonov, Rajesh K. Murarka

**Affiliations:** 1Faculty of Chemistry, University of Gdańsk, Wita Stwosza 63, 80-308 Gdańsk, Poland; cezary.czaplewski@ug.edu.pl (C.C.); adam.sieradzan@ug.edu.pl (A.K.S.); agnieszka.lipska@ug.edu.pl (A.G.L.); sergey.samsonov@ug.edu.pl (S.A.S.); 2Department of Chemistry, Indian Institute of Science Education and Research Bhopal, Bhopal Bypass Road, Bhopal 462066, MP, India; rkm@iiserb.ac.in

**Keywords:** molecular dynamics, coarse graining, biological macromolecules

## Abstract

Molecular dynamics with coarse-grained models is nowadays extensively used to simulate biomolecular systems at large time and size scales, compared to those accessible to all-atom molecular dynamics. In this review article, we describe the physical basis of coarse-grained molecular dynamics, the coarse-grained force fields, the equations of motion and the respective numerical integration algorithms, and selected practical applications of coarse-grained molecular dynamics. We demonstrate that the motion of coarse-grained sites is governed by the potential of mean force and the friction and stochastic forces, resulting from integrating out the secondary degrees of freedom. Consequently, Langevin dynamics is a natural means of describing the motion of a system at the coarse-grained level and the potential of mean force is the physical basis of the coarse-grained force fields. Moreover, the choice of coarse-grained variables and the fact that coarse-grained sites often do not have spherical symmetry implies a non-diagonal inertia tensor. We describe selected coarse-grained models used in molecular dynamics simulations, including the most popular MARTINI model developed by Marrink’s group and the UNICORN model of biological macromolecules developed in our laboratory. We conclude by discussing examples of the application of coarse-grained molecular dynamics to study biologically important processes.

## 1. Introduction

Molecular dynamics (MD) is an extremely useful methodology to investigate the behavior of biomolecules and biomolecular systems and soft matter in general at a level of detail that is still inaccessible during experiments [1,2,3,4,5]. Although, as with all simulation-based methods, it can lead to unwanted artefacts and mis-conclusions, a careful researcher who consistently checks the obtained results against the available experimental data can make important advances in the field with its use. The most accurate treatment is that at the quantum-mechanical level [6,7]; however, this approach is nowadays restricted to small systems. As is commonly understood, MD implies carrying out simulations at the all-atom (AA) level. Owing to tremendous advances in computer techniques and algorithms [3], including the construction of the ANTON supercomputer by the D.E. Shaw group [8,9], millisecond-scale simulations are now feasible for small proteins and it is even possible to run simulations of entire viruses or cell fragments at the microsecond scale [10]. This is, however, still not enough to cover the time scale of biological events. Therefore, 17 years after Alder and Wainwright [1] had published the results of their first MD simulation, the reduction of atomistic representation of the system studies was attempted by Levitt and Warshell [11]. This reduction is termed *coarse graining* (CG). With coarse graining, it is possible to extend the time and size scale of simulations by orders of magnitude, thereby covering the biological scales [12,13,14].

Coarse graining is, however, much more difficult to carry out and has many more caveats compared to the all-atom approaches. First, the degrees of freedom that are not present in a CG model (the fine-grained degrees of freedom) are ignored. This seems to be fine, because a high-school student, who is posed the problem of determining the time after which a brick dropped from height *h* will hit the ground, does not need to consider the atomic details of the brick and needs only to know the “coarse-grained” Newton laws of dynamics. However, the separation of the time scales of the motions of the coarse-grained and the fine-grained degrees of freedom is much more diffuse than that in the physics textbook example mentioned. As a result, the relationship between the time scale of coarse-grained MD simulations and the real or all-atom simulation time scale is largely obtained by comparing the times of events that can easily be traced (e.g., helix formation) or calculated rate constants with those from the experiment or all-atom simulations [15,16,17,18].

A related problem is the construction of the equations of motion. In almost all approaches, the extended sites are represented by point masses, just as atomic nuclei in all-atom simulations [12]. However, this representation of CG sites does not seem to be satisfactory, because these object are, generally, non-spherical. Replacing the full inertia matrix with a diagonal one, composed of point masses, is likely to influence the dynamics. Rigid-body quaternion dynamics [19,20] or treating the sites as stretchable rods [21] are plausible ways to take into account site anisotropy.

The most significant challenge, though, is the development of the coarse-grained force fields [12,13,14,22,23]. Usually, the functional expressions are copied from all-atom force fields, which results in insufficient capacity to model the structures of the systems under study (e.g., protein structures). The reason for this is that the fine-grained degrees of freedom, although not considered in CG models, can give rise to strong coupling between the coarse-grained degrees of freedom [24,25,26]. A variety of approaches have been designed to tackle these problems, including the multiscale force-matching approach [27] and scale-consistent expansion of the potential of mean force into Kubo cluster cumulants [24,25,26]. Nevertheless, the force fields with all-atom-like expressions for energy components can be used to model the dynamics and to estimate the properties of the systems studied, provided that they are applied with care.

The purpose of this article is to provide the reader with an overview of coarse-grained molecular dynamics and its applications in the study of biologically important systems. We do not describe here the approaches based on the Monte Carlo (MC) methodology, some of which, such as the CABS model of proteins [28] and the SimRNA model of nucleic acids [29], are very successful in modeling protein structures, simulating folding pathways, and protein–protein docking [30,31] or simulating RNA folding. We start from the theory, by sketching the derivation of the equations of motion and showing that the effective forces acting on coarse-grained degrees of freedom can be separated into the mean potential forces and the friction and stochastic forces that arise from the motion of the omitted degrees of freedom. Next, we discuss the equations of motion used in different CG MD schemes and the numerical algorithms of solving them. Subsequently, we describe the approaches to coarse-grained force field derivation and discuss selected coarse-grained force fields. Lastly, we discuss the applications of CG MD, illustrating them with examples from our research, and conclude by discussing possible directions of the developments of CG MD approaches.

## 2. Theory and Methodology

### 2.1. Origin of CG Dynamics: Separation of the Coarse-Grained and
Fine-Grained Motions

Let us consider a system, such as a chunk of a polysaccharide chain, which is coarse-grained, by selecting some variables (e.g., coordinates of the glycosidic-oxygen atoms), the other ones being ignored (Figure 1).

Let $R={\left[R_{1},R_{2},\ldots ,R_{M}\right]}^{T}$ and $P={\left[P_{1},P_{2},\ldots ,P_{M}\right]}^{T}$ denote the coarse-grained degrees of freedom and the associated momenta, while $r_{i}={\left[r_{i1},r_{i2},\ldots ,r_{in_{i}}\right]}^{T}$ and $p_{i}={\left[p_{i1},p_{i2},\ldots ,p_{in_{i}}\right]}^{T}$ are the fine-grained degrees of freedom and momenta corresponding to the atoms of site *i*. At this point, we assume that the Rs and the Ps are the coordinates of the centers of mass of the sites and the corresponding momenta. By using the Mori–Zwanzig projection-operator formalism and Liouville equations [15,32,33], Kinjo and Hyodo [16] derived the equations of motion for the coarse-grained degrees of freedom (Equation (Equation 1)). Equation (Equation 1) was derived under the assumption that the coarse-grained degrees of freedom are the mass centers of the coarse-grained sites; however, its extension to generalized coordinates is straightforward.
(1)$$\frac{d}{dt}P_{i}\left(t\right)=-\nabla_{R_{i}}W\left(R\right)-\frac{1}{k_{B}T}\sum_{j}\int_{0}^{t}\langle \left[\delta F_{i}^{Q}\left(\tau \right)\right]{\left[\delta F_{j}{\left(\tau \right)}^{Q}\right]}^{T}\rangle v_{i}\left(\tau \right)d\tau +\delta F_{i}^{Q}\left(t\right)$$
where W(R) is the potential of mean force, $\delta F_{i}^{Q}$ is the fluctuating force (around the mean force value) acting on the *i*th coarse-grained site, $k_{B}$ is the Boltzmann constant, *T* is the absolute temperature, $v_{i}=P_{i}/M_{i}$, where $M_{i}$ is the mass corresponding to the *i*th coarse-grained site, is the velocity of the *i*th coarse-grained site, $\nabla_{R_{i}}=\left(\frac{\partial }{\partial X_{i}},\frac{\partial }{\partial Y_{i}},\frac{\partial }{\partial Z_{i}}\right)$ is the gradient operator in the Cartesian coordinates of the *i*th site, and 〈⋯〉 denotes the ensemble average. The first term on the right-hand side of Equation (Equation 1) is the mean force acting on the *i*th coarse-grained site, the second term corresponds to the friction forces and the last term refers to the fluctuating (stochastic) forces. The potential of mean force and the fluctuating forces are defined by Equations (Equation 2) and (Equation 3), respectively. The potential of mean force will be discussed in Section 2.3.
(2)$$W\left(R\right)=-k_{B}T\ln\frac{\int_{r}\exp\left[-\frac{V(R;r)}{k_{B}T}\right]d\Omega_{r}}{\int_{r}d\Omega_{r}}$$
where V(R;r) is the potential energy function of the system and $d\Omega_{r}$ is the volume element corresponding to the fine-grained degrees of freedom.
(3)$$\delta F_{i}^{Q}\left(t\right)=e^{t\hat{Q}\hat{L}}\delta F_{i}^{Q}\left(0\right)$$
where $\hat{L}$ is the Liouville operator and $\hat{Q}=1-\hat{P}$, where $\hat{P}$ is the projection operator of the coordinates and momenta of the all-atom system to the coarse-grained coordinates and momenta [15,16,32,33].

From Equation (Equation 1), it follows that the net motion of the CG degrees of freedom is governed by the mean forces (the forces averaged over the atoms that constitute the interaction sites), while the fine-grained degrees of freedom contribute to the motion of the CG degrees of freedom through the friction forces (which depend on the time correlation of the fluctuations of the forces acting on the sites) and the fluctuating forces. The friction forces depend on the whole history of the correlation between the fluctuating forces and the velocities of the coarse-grained centers [17,34]. A number of studies were carried out with liquids or simple polymers to determine the friction term exactly based on all-atom simulations. The reader is referred to the excellent review by Klippenstein et al. [17] for details.

In practical implementations, the friction and the stochastic force terms are taken from the Langevin equation [35], which is equivalent to the assumption that the fluctuating forces and the coarse-grained velocities are correlated only over an infinitesimally small period of time (δ-correlated); in other words, it is assumed that the fine-grained degrees of freedom move much faster than the coarse-grained ones. This results in replacing the last two terms on the right-hand side of Equation (Equation 1) with the net friction and stochastic force terms, respectively (Equation (Equation 4)) [15,17,21,34,36,37].
(4)$$G\ddot{q}=-\nabla_{q}U\left(q\right)-\Gamma \dot{q}+f^{rand}$$
where G is the inertia matrix, q, $\dot{q}$ and $\ddot{q}$ are the generalized coarse-grained coordinates, velocities and accelerations, respectively, *U* is the effective coarse-grained energy function (which originates in the potential of mean force; see Section 2.3) [24,25,27], Γ is the friction matrix, and $f^{rand}$ are the random forces. From Equation (Equation 1), it can be inferred that the friction and stochastic force term are interrelated; in fact, they are related through the friction matrix, as given by Equation (Equation 5).
(5)$$f^{rand}={\left(\frac{k_{B}T}{\delta t}\Gamma \right)}^{\frac{1}{2}}N\left(0,1\right)$$
where N(0,1) is the multidimensional normal distribution with zero mean and unit variance–covariance matrix and δt is the time step.

The friction and stochastic forces can be divided into those due to the solvent (the degrees of freedom of which are wholly or in part averaged over in the CG approaches) and those due to the averaged over internal degrees of freedom of the solute, which give rise to the so-called internal friction. In implicit solvent CG models, all solvent degrees of freedom are averaged over. Therefore, the internal friction has a significantly smaller effect on the dynamics than the solvent friction [38]. Consequently, the stochastic and the friction forces are assumed to be due to the solvent. For explicit solvent CG models, the solvent contribution to friction is smaller but still significant, because the rotational degrees of freedom of the solvent molecules are averaged over and, therefore, the stochastic and friction forces should also be included in MD with these models. The solvent viscosity is usually scaled down in MD simulations with implicit solvent CG models and, consequently, the friction and stochastic forces in such simulations are only a fraction of those resulting from Equation (Equation 1), this giving rise to the tremendous speed-up of the CG dynamics with respect to the AA dynamics [37].

In the overdamped limit, when the friction is so high that the left-hand side of Equation (Equation 4) can be ignored, it becomes a system of first-order differential equations that describes the Brownian dynamics.

### 2.2. Implementation of Coarse-Grained MD

Equation (Equation 4) can be applied to any choice of coordinate system. In most of the applications, the coordinates of the centers of masses (CM) of the coarse-grained sites are selected and the sites are treated as point masses [27,39]. In this case, the matrix of inertia is diagonal (with site masses as elements). The friction matrix also is typically diagonal, the coefficients being the net friction coefficients of the interaction sites (Equation (Equation 6)) [3,4]. For implicit solvent CG models, it is assumed that only the solvent contributes to the friction. In some approaches, hydrodynamic interactions with the solvent are considered [40,41,42], the friction matrix having usually the Rotne–Prager form [43].
(6)$$\begin{array}{c} M_{I}{\ddot{R}}_{I}=-\nabla_{R_{I}}U\left(R\right)-\gamma_{I}{\dot{R}}_{I}+\sqrt{\frac{k_{B}T\gamma_{I}}{\delta t}}N\left(0,1\right) \end{array}$$
(7)$$\begin{array}{c} R_{I}=\frac{\sum_{i\in \{I\}}m_{i}r_{i}}{M_{I}} \end{array}$$
(8)$$\begin{array}{c} M_{I}=\sum_{i\in \{I\}}m_{i} \end{array}$$
where $R_{I}$ is the position of the center of the mass of site *I* (with mass $M_{I}$), $\gamma_{I}$ is the friction coefficient of site *I*, N(0,1) is the 3D vector of normally distributed numbers with zero mean and unit variance, and δt is the time step. The friction coefficients are usually expressed by using the Stokes’ law (Equation (Equation 9)) [44].
(9)$$\gamma_{i}=6\pi \rho_{I}\eta$$
where η is the solvent’s viscosity and $\rho_{I}$ is the Stokes’ radius of a site, which is obtained by adding the effective dimension of the water molecule (1.4 Å) to the site’s effective van der Waals radius; the radius is often scaled by the fraction of the solvent-accessible surface of a site. The solvent viscosity is usually scaled down to speed up simulations (typically by a factor of 100 [37]).

The coarse-grained sites have, however, moments of inertia and, therefore, considering only the motion of their centers of masses is often insufficient (a good example is the dynamics of a liquid composed of anisotropic particles). The rotational dynamics of the coarse-grained particles is usually handled in rigid-body mode through the quaternion formalism [19,20,45]. We have developed an alternative formalism, in which the interaction sites are not assumed rigid but to have axial symmetry after averaging out the secondary degrees of freedom [21,36], which is the case of proteins and other biopolymers. The interaction sites are assumed to behave as uniformly stretchable rods, this resulting in a full, albeit constant, inertia matrix, which can be inverted once and its inverse used to obtain accelerations from forces (Equation (Equation 4)). We initially applied this formalism to the UNRES model of polypeptide chains developed in our laboratory [46]; subsequently, we extended it to the Unified Coarse-Grained (UNICORN) model of biological macromolecules, which also encompasses nucleic acids and polysaccharides [47]. The sites are assumed to be linked with virtual bonds, a virtual bond being the axis of a site. The resulting equations of motion can be expressed by Equation (Equation 10) [21,37].
(10)$$\left(A^{T}MA+I\right)\ddot{q}=-\nabla_{q}U\left(q\right)+A^{T}\Gamma \ddot{q}+A^{T}f^{rand}$$
where A is the matrix of the (linear) mapping of the generalized coordinates q to the Cartesian coordinates of the site centers, M is the diagonal matrix of the masses of the sites, I is the diagonal matrix of the moments of inertia of the sites about the axes perpendicular to their virtual bond axes, Γ is the matrix of friction coefficients of the sites (which is usually assumed to be composed of diagonal elements only, but full friction tensors of the sites or the full Rotne–Prager tensor of hydrodynamic interactions can be introduced [42]) and $f^{rand}$ is a vector composed of the vectors of random forces acting on the sites. Initially [21,36], we used the virtual bond vectors as generalized coordinates, this resulting in a full inertia matrix; recently [26,48], we replaced the vectors with the anchor points of the sites (the α-carbon and side-chain-center coordinates in UNRES), thereby reducing the inertia matrix.

Many algorithms have been designed to solve the equations of motion numerically, most of them being based on the Verlet, velocity Verlet and leap-frog schemes [3]. These algorithms are symplectic, i.e., they conserve the total energy, if the right-hand side of the equation of motion contains only the potential force term. For these algorithms, the error in trajectory is proportional to the square of the time step, which seems to be a disadvantage when compared with the higher-order algorithm (e.g., the Runge–Kutta family algorithms or the Gear algorithm [3]) designed to solve the systems of differential equations. However, the energy conservation property, which is not a feature of the more sophisticated algorithms mentioned above, enables us to obtain reliable trajectories even with large time steps, especially when the multiple-time-step or time-split algorithms are applied, in which the Liouville operator is split into the part corresponding to fast- and slow-varying forces [3]. With the time-split feature, it is possible to extend the time step even to 10 fs [21].

With the friction and stochastic forces, not the total energy but the kinetic temperature (defined by Equation (Equation 11)) is to be conserved; these contributions to forces provide, therefore, a thermostat.
(11)$$T_{kin}=\frac{E_{k}}{N_{f}k_{B}}$$
where $E_{k}$ is the kinetic energy of the system and $N_{f}$ is the number of the degrees of freedom. The Langevin thermostat is very stable and provides correct temperature distribution [49]. Therefore, its use has been recommended even in all-atom MD with explicit solvent [50]. In this case, the non-conservative forces can be considered as those due to the environment of the system that is subjected to MD simulations.

Inclusion of the friction and stochastic forces in numerical integration of the equations of motion is not straightforward. In the simplest instance, they can be included explicitly and this is the method of choice when the friction matrix is not diagonal [37]. However, the stochastic forces vary rapidly, which can result in problems with temperature conservation when the time step is large. The most elaborate algorithm has been developed by Ciccoti and coworkers [51,52], in which the friction and stochastic forces are pre-integrated over the time step, this resulting in exponential terms in the Verlet-family integrators. However, this algorithm becomes prohibitively expensive with the full friction matrix, because calculating the matrix exponential requires its diagonalization. Thermostating can also be achieved without the stochastic and friction term through the use of the Berendsen [53,54], Nosé–Hoover [55] or Nosé–Poincaré [56] thermostats; the latter two also enable us to rigorously control temperature conservation [52].

Apart from full-blown equations of motion, also the discontinuous MD, which is reminiscent of the early work of Alder and Wainwright [1], has been implemented with some CG models. An example is the knowledge-based PRIME 20 CG model of polypeptide chains [57,58].

### 2.3. Effective Potential Energy Functions

As demonstrated in Section 2.1, the mean forces acting on the coarse-grained degrees of freedom (the negative of the gradient of the potential of mean force) are the conservative forces that govern the motion of the system in coarse-grained representation. A natural consequence of this observation is that the prototype of the effective energy function of a coarse-grained system is the potential of mean force, in which the secondary degrees of freedom are integrated out (Equation (Equation 2)). This definition is, indeed, applied to derive the physics-based force fields [24,27,39] and can even be extended to derive the statistical force fields, the components of which are obtained by Boltzmann inversion of the distribution functions extracted from structural databases [59,60,61]. However, the exact PMF is specific for a given system and prohibitively expensive to determine. Therefore, approximations have to be developed, in which the effective energy function is split into transferable components, as in all-atom force fields. This approach was originated by the seminal work by Levitt and Warshel [11], who proposed the first physics-based coarse-grained model of proteins and derived the CG terms by Boltzmann averaging out the all-atom energy of model systems. The two major approaches are the bottom-up and the top-down approach [12,26,39]. In the first one, the energy terms are obtained from atomistically detailed interactions, as in the work by Levitt and Warshell [11], while in the top-down approach, a force field is constructed so as to reproduce the measurable properties. A number of methods combine the bottom-up approach in the derivation of coarse-grained energy terms and the top-down approach to put them together into a working force field. In this section, we summarize some of the coarse-grained force fields and methods of their derivation, referring the reader to dedicated review articles [23,61,62,63,64] and books [12,13,14] for details.

The most straightforward method of the construction of a coarse-grained force field is to import the respective expressions from the all-atom force fields. The effective energy function is then a sum of local terms, corresponding to virtual bond stretching ($U_{s}$), virtual bond angle bending ($U_{\theta }$), torsional terms corresponding to the rotation about the virtual bonds ($U_{tor}$) and the pairwise interaction terms that consist of the electrostatic (usually Coulombic) and nonbonded terms, the latter usually expressed by the Lennard–Jones potential, which can be modified to introduce a softer repulsion term or dependence on site orientation (the most commonly used form of the latter is the Gay–Berne potential [65]). The solvent is either treated explicitly (usually by coarse graining several water molecules into one center) or is implicit in the potentials; in the second case, an additional term $V_{solv}$ can be added to account for, e.g., exposure of a site to the solvent. The whole energy expression is given by Equation (Equation 12). Expressions of this type are referred to as the neoclassical expressions. Such expressions are implemented in the most popular MARTINI force field [66,67,68,69,70].
(12)$$\begin{array}{c} U=\sum_{i}\frac{1}{2}k_{i}^{d}{\left(d_{i}-d_{i}^{\circ }\right)}^{2}+\sum_{i}\frac{1}{2}k_{i}^{\theta }{\left(\theta_{i}-\theta_{i}^{\circ }\right)}^{2}+\sum_{i}\sum_{n}a_{i}^{\left(n\right)}\left[1+\cos(n\gamma_{i})\right]+b_{i}^{\left(n\right)}\left[1+\sin(n\gamma_{i})\right]+\\ 332\sum_{i<j}\frac{q_{i}q_{j}}{Dr_{ij}}+4\varepsilon_{ij}\left[{\left(\frac{\sigma_{ij}}{r_{ij}}\right)}^{12}-{\left(\frac{\sigma_{ij}}{r_{ij}}\right)}^{6}\right]+V_{solv} \end{array}$$
where $d_{i}$, $d_{i}^{\circ }$ and $k_{i}^{d}$ are the length, equilibrium length and force constant of the *i*th virtual bond, $\theta_{i}$, $\theta_{i}^{\circ }$ and $k_{i}^{\theta }$ are the actual and equilibrium value and the force constant of the *i*th virtual bond angle, $\gamma_{i}$ is the *i*th virtual bond dihedral angle, $a_{i}^{\left(n\right)}$ and $b_{i}^{\left(n\right)}$ being the coefficients in the expressions for the torsional potentials, $q_{i}$ is the partial charge on the *i*th site, *D* is the relative dielectric permittivity, $\sigma_{ij}$ and $\epsilon_{ij}$ are the constants of the Lennard–Jones potential for the interaction of site *i* with site *j*, and $r_{ij}$ is the distance between these two sites. The coefficient of 332 in the expression for the Coulombic energy is introduced to express the energy in kcal/mol if the distance is expressed in ångstroms and the charges are expressed in electron charge units.

The neoclassical expressions usually do not produce the force fields that can model the structures of the systems under study without external information. The reason is that the PMF cannot be broken down into pairwise terms, even if the parent all-atom energy function is pairwise [24,25,60]. Multibody terms need to be introduced for structure modeling. Moreover, the site–site interaction potentials of most coarse-grained force fields are usually too “sticky”, this resulting in too compact modeled structures [71,72,73,74,75]. The “stickiness” arises from interaction imbalance, which can be caused by the absence of specific cross Lennard–Jones parameters for pairs of particles of different size or overly weak bond force constants [76]. This problem was resolved recently in the MARTINI 3 force field [77].

A method for the derivation of explicit multibody terms, which is based on the expansion of the PMF into Kubo cluster cumulant functions [78], has been developed in our laboratory [24,25]. Following this method, the PMF is expressed as a sum of elementary PMFs characteristic of single sites, pairs of sites and higher clusters of sites. The factors can be identified with single-body, pairwise and multibody terms in the effective energy function. Each factor can be expanded into the Kubo generalized cumulants [78] and the lowest-order cumulants can be used as templates of the analytical expressions for the respective energy terms. Notably, the cumulant-based derivation of the effective energy terms enables us to determine which effective energy terms should be scaled depending on temperature [79]; that this should be the case is easy to realize given the origin of the CG energy functions in the PMF (Equation (Equation 2)). Recently [25,26], we developed a generalized mathematical formalism for the derivation of the formulas for the cumulants for the coarse-grained representations of polymer chains, by expanding the all-atom energy in the trigonometric functions of the collective rotation angles of the groups about the virtual bond axes (Figure 1). The obtained formulas implicitly include the atomic details of the system; in particular, they keep the correct dependence of the interaction energy of the non-radial sites on orientation and the correct dependence of the site–site interaction energy on local geometry, through multibody terms. We have termed this approach the scale consistent approach to force field derivation. The procedure has been summarized and illustrated with examples in our recent review article [26]. This feature solves part of the problem of force field “stickiness” pointed out in [75], because the force field contains explicit terms that couple the backbone-local and backbone-electrostatic conformational states, which prevent overly compact local chain fragments [80]. Another way of including the multibody terms is to introduce expressions derived based on structural regularities in crystal structures [60,81] or on a heuristic basis [82,83].

As mentioned, the parametrization of the effective energy function follows the bottom-up, the top-down approach or a combination of both. In the bottom-up approach, the individual components are parametrized based on direct numerical integration of all-atom energy surfaces [11,84,85,86] or from MD simulations of model systems [87,88,89]. In the top-down approach, all parameters are optimized simultaneously. One method is to reproduce the physicochemical properties such as the partition coefficients between the aqueous and lipid phase, the structural properties (e.g., the virtual bond and virtual bond dihedral angle distribution from the Protein Data Bank (PDB)) or the observables obtained from the all-atom MD simulations. This approach has been used to develop the MARTINI force field [66,67]. A systematic feedback between the CG and the corresponding AA system with iterative improvement of the parameters is achieved through the iterative Boltzmann inversion [90,91] and inverse Monte Carlo (IMC) method [92,93]. A more sophisticated approach is the relative entropy minimization [94,95,96,97], in which the Kullback–Leibler divergence [98] between the CG and AA ensembles is minimized. We developed a variant of this method, which we termed maximum-likelihood optimization [99], replacing the AA-simulated ensembles with the experimental ensembles, obtained from Nuclear Magnetic Resonance (NMR) studies of the training proteins at various temperatures, which were selected to bracket the folding–unfolding transition [100]. Including the experimental ensemble required “Gaussian smearing” of each of the experimental conformations so that it could be matched to the conformation of the simulated ensemble. This method was implemented in optimizing the UNRES force field for proteins [80,101] and the NARES-2P force field for nucleic acids [102].

Because one of the purposes of running MD simulations is obtaining information about the dynamic behavior of a system, not only the compatibility between the CG effective energy surfaces with the AA potential energy surfaces but also the compatibility of the CG forces with the average forces computed at the AA level is very important. To achieve this, the force-matching method can be used, which has been developed for the purpose of parametrizing the CG force field by the Voth group [27,39,103] as the Multi-Scale Coarse-Grained (MS-CG) method. In this method, the CG energy function is parametrized to minimize the sum of the squares of the differences between the CG and the corresponding average AA forces. It is assumed that the centers of the masses of the sites are coarse-grained centers, all interactions have radial symmetry, and all effective interactions are pairwise. Consequently, the distance profiles of the site–site potentials can be determined without assuming any specific functional form. The multibody terms are assumed to be embedded into the effective potentials. We extended the method to non-spherical and multibody potentials (however, assuming their functional forms) and recently used it to parametrize the UNRES force field, obtaining its variant that produces forces compatible with those computed with the all-atom approach; it is transferable and is able to fold proteins [26].

Machine learning is playing an increasing role in molecular modeling, the most spectacular success being achieved by Google Deep Mind in the 14th Community Wide Experiment on the Critical Assessment of Techniques for Protein Structure Prediction (CASP14), in which the group predicted the structures of most of the target proteins with crystallographic accuracy [104,105]. It is also increasingly used in force field optimization, regarding both the all-atom [106,107,108] and coarse-grained force fields [109,110,111].

Selected CG force fields used in MD simulations are summarized in Section 2.3.1, Section 2.3.2, Section 2.3.3, Section 2.3.4, Section 2.3.5, Section 2.3.6 and Section 2.3.7. Section 2.3.7 is devoted to glycosoaminoglycans, which, despite their significance in cell functioning, seem to be underrepresented in the literature. Apart from these force fields, MD simulations are carried out with the structure-based or Gō-like models [112,113], which are designed so that the interactions between the residues that are in contact in the experimental structure possess energy minima, while the other interactions are all repulsive, and the elastic network models, in which harmonic, anharmonic or double-well potentials are employed to keep the structure in the neighborhood of the native structure [114,115,116]. The Gō-like models have been used to study protein folding [117,118,119], including the folding and unfolding of knotted proteins [120], and to study the mechanostability of virus capsids [121]. The elastic network models are used in investigations of the fluctuations around the native structures, particularly in finding the functionally important motions [114,115,116].

#### 2.3.1. AWSEM

The Associative memory, Water mediated, Structure and Energy Model (AWSEM) [122] has been developed in the Wolynes and Papoian labs to handle protein systems. The geometry of the chain is defined in terms of C${}^{\alpha }$, C${}^{\beta }$ and backbone carboxyl-oxygen atoms, which are interaction sites; the positions of backbone N and H atoms are calculated assuming the ideal trans-peptide group geometry. The energy function is a sum of the backbone potential, which consists of the connectivity, Ramachandran, chirality and excluded volume terms, the contact potential that consists of the pairwise C${}^{\beta }\cdots$C${}^{\beta }$ (C${}^{\beta }$ replaced with C${}^{\alpha }$ for Gly) and the water-mediated potentials, the burial potential, the hydrogen-bonding potential that depends on the oxygen–nitrogen and oxygen–hydrogen distances, peptide group orientation and an explicit helical term, the associative memory term, which encodes the alignment to proteins with known structures, and the desolvation term. AWSEM has been integrated into the LAMMPS [123] general MD package. It has been used successfully in protein structure prediction and simulations of protein folding and assembly [124], including protein aggregation [125].

#### 2.3.2. MARTINI

MARTINI [69,76,77] is the most popular CG force field. Originally developed to simulate lipid systems [66], it has been extended to proteins [67], polysaccharides [68], nucleic acids [70] and materials science [126]. A major advantage of MARTINI is a standardized coarse-graining scheme, in which chain fragments are merged into sites comprising four non-hydrogen atoms on average, while rings are divided into three-atom fragments. Depending on the character, each CG particle is assigned a polar (P), nonpolar (N), apolar (C) or charged (Q) type, with standardized parameters. This scheme enables automatic coarse-graining without user intervention. Water and lipid molecules are explicit, except for the variant termed dry MARTINI [127]. Four water molecules are assembled into an extended solvent site. The energy function is of neoclassical type (Equation (Equation 12)). MARTINI was originally developed to work with the GROMACS MD suite [128] but is also used with other standard MD packages.

MARTINI has been used worldwide to simulate a variety of biological systems [69]. It has also been successfully applied in scoring the docked protein structures [129] and in protein–RNA docking [130]. However, the force field cannot predict protein secondary structure and, consequently, secondary structure restraints have to be imposed when simulating systems containing proteins and nucleic acids.

#### 2.3.3. OPEP and HiRe-RNA

Optimized Potential for Efficient protein structure Prediction (OPEP) [82,83,131] is a CG force field, which was initially developed to perform simulations of polypeptides. As in AWSEM, the all-atom representation of the polypeptide backbone is used, while each sidechain is represented by a single spherical bead. The effective energy function consists of local (bond stretching, bond angle, torsional) and long-range terms, which include the sidechain–sidechain contact potentials and a sophisticated backbone-hydrogen-bonding term, which reflect the nearly linear arrangement of the N-H⋯O groups as observed in experimental structures. The H-bond potentials also include four-body terms that promote the formation of regular hydrogen bond patterns. The force field has been parametrized using a combination of bottom-up and top-down approaches. Recently [132], the model has been extended to run constant pH simulations, which is accomplished by Monte Carlo sampling of protonation states. Owing to careful design and parametrization, unrestrained folding simulations can be performed with OPEP. Based on OPEP, the PEP-FOLD3 server [133] was created, which enables the user to run peptide and small protein folding simulations.

Based on the concept of OPEP, the HiRe-RNA model of nucleic acids [134] was developed, in which four interaction sites (P, O5’, C5’, C4’ and C1’) are assigned per backbone unit, a pyrimidine base is represented by one bead and a purine base is represented by two beads. A special orientation-dependent potential has been designed to keep paired nucleic acid bases at appropriate geometry. OPEP/HiRe-RNA have been used in investigating protein dynamics, including the effect of hydrodynamics interactions, small protein and RNA folding, and in investigating protein aggregation, including amyloid formation [131]. The OPEP force field, after training, was able to score structures of protein–protein complexes with higher accuracy than that of ZDOCK [135].

#### 2.3.4. oxDNA and oxRNA

oxDNA/oxRNA, developed by Ouldridge, Louis and Doye [136,137,138,139,140,141], is a low-resolution DNA and RNA model with three beads per nucleotide. These sites are the backbone repulsion site, the base repulsion site and the base hydrogen-bonding/stacking site, respectively, and they form a linear fragment perpendicular to the direction of the chain. The interaction potentials have been engineered to reproduce base pairing but depend on the kinds of interacting bases and not on their position in the sequence. Salt effects have been included [140]. The model has been parametrized to reproduce the geometry and thermodynamics of DNA hybridization and has been used with success to model the formation and rearrangements of DNA and RNA nanostructures [141]. The model has been included in the oxDNA package [140].

#### 2.3.5. SIRAH

The Southamerican Initiative for a Rapid and Accurate Hamiltonian (SIRAH) force field [142,143] is a CG force field to treat proteins, later extended to nucleic acids [144], lipids and polysaccharides. There are three interaction sites per residue for the polypeptide backbone, located at the amide-N (GN), C${}^{\alpha }$ (GC) and carbonyl-O (GO) atoms, while each sidechain is represented by 1 (for Gly) to 7 (for Arg and Trp) beads. Water is explicit in the model. Each “water” particle (WT4) represents 11 tetrahedrally coordinated water molecules and comprises four beads. Hydrated ions Na${}^{+}$, K${}^{+}$ and Cl${}^{-}$ are considered explicitly, each extended ion particle comprising the actual ion and the six water molecules constituting the first hydration shell. As in OPEP, the interaction potential consists of local and long-range terms. Coulombic interactions are calculated explicitly and partial charges are present on the beads of the WT4 particles, which results in the appropriate handling of hydration. The parameters have been determined to reproduce the structural and thermodynamic properties of model systems. SIRAH has been ported to GROMACS [128] and AMBER [145]. It has been used in the simulation of natively unfolded proteins and protein aggregation.

#### 2.3.6. UNICORN

The UNIfied COarse gRaiNed model (UNICORN) [26,47] is the most heavily coarse-grained model of those described in this section. It has been created by merging the models specific for proteins (UNRES) [24,47,80,146], nucleic acids (NARES-2P) [147] and for polysaccharides (SUGRES-1P) [47,148,149]. Recently, its extension to treat protein–nucleic acid complexes has been developed [150]. For each macromolecule type, one bead is placed in the middle of the backbone virtual bond to represent a peptide group, a phosphate group or a sugar ring, respectively, and one on the side group, if applicable. The respective potentials have been derived by using the recently developed scale-consistent theory [25,26], which was evolving together with UNRES [24,25,151]. The energy function consists of local (virtual bond stretching, virtual bond angle bending, virtual bond torsional, sidechain chirality) and long-range terms, including the multibody terms. The torsional terms depend both on virtual bond dihedral angles and on the virtual bond angles, which is a consequence of the application of the scale-consistent theory [25]. Owing to careful physics-based derivation of the energy function, UNICORN can model regular secondary structure patterns without engineering specific potentials. Solvent is implicit in the interaction potentials. The force field has recently been extended to the lipid bilayer environment, which is treated as a continuous medium [152]. The force fields have been calibrated with a combination of the bottom-up and top-down approaches [25,47,80,102,147]. UNICORN is probably the only coarse-grained model in which the effective energy function depends on temperature [79].

UNRES is applicable to the energy-based prediction of protein structures [153,154,155,156,157,158], to study protein folding [159] and free-energy landscapes [160] and to solve a variety of biological problems [47,161,162], including the formation of oligomers and fibrils of amyloidogenic peptides [163,164,165,166]. Some of these studies are briefly described in Section 3. UNRES also enables the researchers to run simulations of proteins that include D-amino-acid [80,167] and phosphorylated residues [168] and to consider the dynamic breaking and formation of disulfide bonds [169]. An extension to treat the binding of proteins to nanoparticles has also been developed [170]. The package is available in standalone form at www.unres.pl (accessed on 10 September 2021) and also through the web server [171], which has recently been extended to add the peptide–protein and protein–protein docking functionality [172].

NARES-2P not only reproduces the double-helix structure and folding thermodynamics of DNA and RNA molecules but also the pre-melting transition in DNA [147]. With limited restraints, NARES-2P is capable of modeling complex DNA and RNA folds [173]. NARES-2P was used to investigate telomere stability [174] and the influence of single-strand breaks on the mechanical stability of various DNA chains [175].

SUGRES-1P is still under development [148,149]. It is being extended [149] to include glycosaminoglycans (GAGs), for which the CG force fields are scarce despite their high biological significance. The other force fields developed for glycosoaminoglycans are described in Section 2.3.7.

#### 2.3.7. Coarse-Grained Potentials for Glycosoaminoglycans

Glycosaminoglycans (GAGs) are a special class of linear anionic polysaccharides composed of disaccharide periodic units that contain a uronic acid and an N-acetylated aminosugar [176] residue, with some of the hydroxyl groups sulfated. The types of particular monosaccharide building blocks, glycosidic linkages and positions of sulfation (“sulfation code” [177]) contribute to the high chemical and structural variety of this family of carbohydrates, which are traditionally classified into several groups: heparin, heparan sulfate, chondrotin sulfate, dermatan sulfate, keratan sulfate. The GAGs are located in the extracellular matrix of the cell, where they participate in a number of biologically relevant processes, such as cell signaling, cell proliferation, cell adhesion, anticoagulation and antiogenesis, by establishing interactions, their targets being collagen [178], growth factors [179], chemokines [180] and cathepsins [181], among others. While all-atom approaches are sufficient to model short GAGs (6–8 units) and their interactions [182], CG models are required for modeling very long GAGs, which contain hundreds and thousands of monosaccharide blocks. Such long GAGs are present in the extracellular matrix [183] and guide processes such as the creation of the protein gradients [184] or establishing collagen networks [180].

The first CG model of GAGs, along with the respective force field, for use in MC simulations, was developed by Bathe et al. [185]. The model contains five centers per GAG unit: the mass center, the charge center and one oxygen and two carbon atoms, forming a glycosidic linkage. It was parametrized based on all-atom MD simulations of model systems. The first model applicable in MD simulations was developed by the Almond group [186,187], which covers heparan sulfate, hyaluronic acid, chodroitin sulfate and dermatan sulfate. In this model, a GAG unit consists of two centers, which are the sugar-ring center and the glycosidic-oxygen atom, respectively. Sugar-ring puckering is also accounted for. The parameters were derived from microsecond all-atom simulations using the GLYCAM06 force field [188]. The model was applied in studies of heparan sulfate polysaccharides consisting of 50–200 units [186] and of proteoglycans consisting of GAGs as a glyco-part at a CG level [187]. The resulting force field is compatible with the GLYCAM06 and AMBER FF03 [189] force fields, thus enabling multiscale simulations of glycans to be carried out.

A more detailed CG representation developed to model 17 different GAGs, including the natural ones (heparin, desulfated heparan sulfate, chondroitin sulfate, dermatan sulfate and hyaluronic acid) and their artificially sulfated derivatives, was proposed by Samsonov et al. [190]. In addition to the sugar-ring center and glycosidic-oxygen atom, N-acetyl, N-sulfate, 4-, 6-sulfates (for aminosugars) and 2-, 3-sulfates (for uronic acids) as well as carboxyl groups (distinguishable for glucuronic and iduronic acids) were introduced as separate centers. The model has been designed for use with the AMBER package and the parameters of the respective force field were obtained from the results of all-atom MD simulations. The model shows good agreement among end-to-end distance and radii of gyration for GAGs with a length of up to around 32 units.

### 2.4. Extensions of MD

Although the original purpose of MD was to study the time evolution of molecular systems, the MD simulations carried out in canonical mode can be used to obtain equilibrium ensembles and, thereby, equilibrium structural averages (e.g., the end-to-end distance), thermodynamics properties (e.g., average energy, heat capacity, etc.) or distributions, which can be compared with the experimental observables. However, because a simulation run at a constant temperature can get stuck in a local minimum region, extensions of MD aimed at enhancing the conformational sampling were developed [46,191,192,193,194,195], which can be divided into three groups. In the methods of the first group, which consist, among others, of umbrella sampling (US) [196,197], thermodynamic integration (TI) [198] and steered molecular dynamics (SMD) [199], the system is guided along a given reaction coordinate. These methods are often combined with the generalized ensemble methods to ensure sufficient sampling. The methods of the second group are metadynamics methods [200,201], in which the system is pushed away from the already visited region(s) of the conformational space, by adding a series of Gaussian terms centered at the encountered values of the selected reaction coordinate. These methods are also often used together with generalized ensemble methods. An approach related to metadynamics, termed Gaussian-accelerated MD, which does not require reaction coordinate selection, was proposed recently [195]. The methods of the third groups are aimed at visiting the whole energy space. This group comprises simulated annealing (SA) [202] and generalized ensemble methods [203,204,205,206]. Aside from the methods summarized above, the forward-flux sampling method has been developed [207,208] for the efficient determination of kinetics from MD.

The generalized ensemble methods are the most efficient in searching the conformational space. Among them, the Replica Exchange Molecular Dynamics (REMD) [203,204] and its variants are the most straightforward and, at the same time, very efficient. In REMD, several MD simulations (replicas) are carried out at different temperatures ($T_{0},T_{1},\cdots ,T_{M_{T}}$). The replicas evolve independently and, after a certain number of MD steps, an exchange of temperatures between neighboring replicas (with indices *i* and i+1, respectively) is attempted, the acceptance of the exchange following the Metropolis criterion with the exchange probability ω expressed by Equation (Equation 13), which reflects the fact that the effective CG energy function can depend on temperature [79,209]:(13)$$\omega \left(q_{i}\to q_{j}\right)=\min\left[1,\exp\left(-\Delta \right)\right]$$
with
(14)$$\Delta =\left[\beta_{j}U\left(q_{j};\beta_{j}\right)-\beta_{i}U\left(q_{j};\beta_{i}\right)\right]-\left[\beta_{j}U\left(q_{i};\beta_{j}\right)-\beta_{i}U\left(q_{i};\beta_{i}\right)\right]$$
where $\beta_{i}=1/k_{B}T_{i}$, $T_{i}$ being the absolute temperature corresponding to the *i*th trajectory, and $q_{i}$ denotes the variables of the conformation of the *i*th trajectory at the attempted exchange point. To enhance the sampling further, several trajectories can be run at a given temperature, this approach being called the Multiplexed Replica Exchange Molecular Dynamics (MREMD) [210,211].

Further enhancement of sampling can be achieved by extending the replica scheme to include the components of the energy function; this extension has been coined the Hamiltonian Replica Exchange Molecular Dynamics (HREMD) [212,213,214]. Thus, a number (e.g., *M*) of canonical MD simulations are carried out simultaneously at different temperatures and with different potential energy functions (*V*), which can differ by the repulsive Lennard–Jones terms, if the purpose is to allow the system to overcome sterical clashes or by the restraint function(s), e.g., umbrella or experimental-restraint potentials, if the purpose is to explore different ranges of order parameters [79] or to perform data-assisted structure modeling [214]. The replicas constitute a two-dimensional ($T_{i},V_{j}$) grid. For each replica, an exchange is attempted with its neighboring replica in one or two dimensions (up, down or diagonal on the grid, the direction being selected at random). The exchange is accepted based on the probability ω expressed by Equation (Equation 15):(15)$$\omega \left(q_{ij}\to q_{kl}\right)=\min\left[1,\exp\left(-\Delta \right)\right],\;\;\left(k,l\right)\in \left\{\left(i+1,j\right),\left(i,j+1\right),\left(i+1,j+1\right)\right\}$$
with
(16)$$\Delta =\left[\beta_{k}V_{l}\left(q_{kl};\beta_{k}\right)-\beta_{i}V_{j}\left(q_{kl};\beta_{i}\right)\right]-\left[\beta_{k}V_{l}\left(q_{ij};\beta_{k}\right)-\beta_{i}V_{j}\left(q_{ij};\beta_{i}\right)\right]$$
where $V_{j}$ is the potential energy function (including the restraining potential) corresponding to the $\left(T_{i},V_{j}\right)$ trajectory, and $q_{ij}$ denotes the variables of the respective conformation of this trajectory at the attempted exchange point. A variant of HREMD was developed recently [215] for mixed-resolution MD simulations with the MARTINI [66] and GROMOS [216] force fields.

Multicanonical algorithms [217,218], also known as entropic sampling [218], are another class of generalized ensemble methods, in which the energy is replaced in the Metropolis criterion by the logarithm of the density of states (the microcanonical entropy). A simulation is converged when all energies are sampled with the same frequency; therefore, this method is well suited to overcome energy barriers. Once the density of states is obtained, all ensemble averages can be computed. The multicanonical algorithms have also been combined with REMD to form the Replica Exchange Multicanonical Algorithm (REMUCA) and Multicanonical Replica Exchange Algorithm (MUCAREM) [204]. These algorithms were implemented in the UNRES CG force field [219].

## 3. Examples

In this section, examples of CG MD studies with UNRES and NARES-2P are discussed, which pertain to investigating the kinetics of protein folding and ensemble-based structure modeling.

### 3.1. Investigation of Protein-Folding Kinetics and Pathways

#### 3.1.1. Folding Kinetics of FBP WW Domain and Its Mutants

The most straightforward comparison of the time scale of coarse-grained simulations with the experimental time scale is that of the timing of the events, which can be captured both by experiments and simulations. The quantities that can be obtained from both experiments and simulations are the rate constants of protein folding. We studied the folding of the Formin Binding 28 (FBP28) WW domain by using UNRES MD [159]. The native structure of this protein is an antiparallel three-stranded β-sheet (PDB: 1E0L) [220] and the mechanism, thermodynamics and kinetics of its folding and of the folding of its mutants were the subject of extensive experimental studies [221]. We carried out a canonical simulation study of the wild-type protein and its three single-residue mutants, Y11R, Y19L and W30F, and three mutants with deletions of five N-terminal or four C-terminal residues or both, ΔNY11R, ΔNΔCY11R and ΔNΔCY11R/L26A. These mutants exhibit different folding rates and stability compared to the wild-type protein [221].

For each protein, 512 canonical simulation trajectories were run at a temperature below the folding transition temperature. The duration of each simulation was at least 5 μs of coarse-grained MD time. The fractions of folded and intermediate structures, defined based on the Root Mean Square Deviation (RMSD) cut-off from the respective experimental structures, were calculated every 2000 MD steps (each with length of 4.89 fs) as averages over the 512 trajectories and single- and double-exponential kinetic equations were fitted to them. Except for the Y19L and Y30F mutants, all proteins were found to exhibit double-exponential kinetics, indicating the presence of an intermediate. This result was in fair agreement with the experiment, which suggests that Y19L and ΔNY11R exhibit a single-exponential and the other protein exhibits a double and single-exponential kinetics, respectively. However, an analysis of the free-energy landscapes demonstrated that the intermediate, which has the N-terminal hairpin fully folded and the C-terminal hairpin partially formed (Figure 2), is always present but, for Y19L, the transition from the intermediate state to the native structure corresponds to a low free-energy barrier, making the kinetics effectively single-exponential.

We compared the rate constants corresponding to the fast phase of folding (no accurate experimental data were available for the slow-phase rate constants; this phase probably corresponds to the formation of the C-terminal β-hairpin [222]). A plot of the correlation between the logarithms of the rate constants determined from simulation and the experimental rate constants is shown in Figure 3. It can be seen that the correlation is good, if only the wild-type protein and single-point mutants are considered, but it deteriorates when the mutants with N- and C-terminus deletions are added. Nevertheless, Figure 3 demonstrates that the experimental rate constants are roughly three orders of magnitude smaller compared to those resulting from simulations, this demonstrating the time-scale extension of CG simulations.

#### 3.1.2. Effect of Hydrodynamic Interactions on Folding

Because our study on the FBP WW domain and its mutants demonstrated that protein-folding kinetics can be modeled reasonably by using MD with UNRES, we undertook an investigation of the effect of the non-diagonal friction tensor (see Section 2.1) due to the solvent (which is implicit in UNRES), which gives rise to the so-called hydrodynamic interactions (HIs), on the folding dynamics. The HIs are manifested as the apparent drag of two objects moving through a liquid [40,41,223]. The friction matrix in hydrodynamic interactions is usually expressed by the Rotne–Prager (RP) tensor [43]. The HIs probably play an important role in protein-folding protein–protein association [224], the formation of lipid membranes [225], the sliding motion of protein along DNA [226], etc.

The studies carried out with the Gō-like models [40,41] suggested that HIs accelerate folding, owing to the faster initial collapse that takes place if they are introduced. However, the Gō-like models are biased towards native interactions. Therefore, we studied the effect of hydrodynamic interactions on the folding kinetics with UNRES, taking the N-terminal domain of staphylococcal protein A (PDB: 1BDD) and the FBP 28 WW domain [42] as examples. The first protein is a 46-residue three-α-helix bundle and, as mentioned above, the second one is a 37-residue three-stranded anti-parallel β-sheet, PDB: 1E0L). For both proteins, we ran two series of 200 independent canonical MD trajectories, one with plain-Langevin dynamics and another one with Langevin dynamics including hydrodynamic interactions, at T=300 K and T=335 K for 1BDD and 1E0L, respectively, which are below the respective transition midpoints equal to 320 K and 339 K for 1BDD and 1E0L, respectively. The progress of folding was monitored as the variation of the fraction of native-like conformations and that of the conformation of the respective intermediate with time, computed from all trajectories at a given snapshot [42].

As opposed to the results of earlier studies [40,41], we found that the folding becomes slower upon introducing hydrodynamic interactions [42] of both proteins. The reason for this is the presence of non-native intermediates (Figure 4), which are kinetic traps. For 1BDD, the intermediate is a mirror image of the native three-helix bundle, and for 1E0L, it has a partially helical conformation. The intermediates persist for a much longer time when hydrodynamic interactions are included and, for 1E0L, even a steady state is reached.

The kinetics was found to be biexponential, with the fast route corresponding to direct folding and the slow one involving an intermediate [42]. The fast route was found to be faster in the presence of hydrodynamic interactions, which led to faster collapse, consistent with the results obtained with the Gō-like models [40,41]. However, the collapse can also lead to an intermediate. The intermediates, especially that in the simulated folding of the FBP WW domain with HIs included, are clearly far from the corresponding native structures, while having a well-defined secondary and tertiary structure, this resulting in a high free-energy barrier of their conversion to the native structures. It should be noted that non-native interactions are not present in the Gō-like Hamiltonians, which is the reason that studies with these models suggested that hydrodynamic interactions accelerate folding.

#### 3.1.3. Phosphorylation-Induced Folding of the Intrinsically Disordered
eIF4E-Binding Protein Isoform 2

The Eukaryotic Translation Initiation Factor 4E (eIF4E)-binding protein isoform 2 (4E-BP2) is a natively unfolded protein that can, however, form a marginally stable four-stranded β-sheet upon post-translational phosphorylation of two threonine residues, the respective variant being pT37pT46 4E-BP2 (PDB: 2MX4) [227]. Phosphorylation also results in an approximately 100-fold decrease in the binding activity. In order to determine the origin of the formation of the structure and the respective folding pathways, we used the recently developed variant of UNRES that includes phosphorylated residues [168].

We ran canonical simulations (80 trajectories) for both the wild type of 4E-BP2${}_{18-62}$ and its pT37pT46 variant. The wild type formed a three-stranded anti-parallel β-sheet in part of the structure but the N-terminal strand remained loose (Figure 5). It turned out that the formation of electrostatic contacts involving the phosphorylated residues is crucial for directing the N-terminal strand to close the complete β-structure. Two types of folding trajectories were found, which are shown in Figure 6. The results of this study demonstrated that coarse-grained modeling with UNRES is able to handle well phosphorylated proteins, which is very important in view of the fact that such proteins are involved in signaling.

### 3.2. Ensemble-Based Modeling of Protein Structures

The structure of a biomolecule or that of a biomolecular system can be understood as the most probable conformational ensemble at a given temperature and other condition. This definition is consistent with Anfinsen’s thermodynamic hypothesis [228] and covers both proteins and other biological macromolecules with a well-defined structure, in which case the ensemble is tight, and natively unfolded proteins. All-atom molecular dynamics is often used to refine all-atom structures [229] and canonical or extended MD and MC sampling is used with various degrees of success in structural modeling [28,47,82,133,230]. With canonical simulations, it is straightforward to perform cluster analysis of the resulting ensemble and select the mean structures from the clusters as candidate models, ranking them according to decreasing cluster populations; this approach has also been carried over to extended canonical ensemble simulations [231]. We have developed a fully energy-based method of protein structure modeling [79,156], which we used in the CASP exercises [155,156,157,158,232,233]. The method consists of four basic stages. In stage 1, an extensive conformational search is carried out with MREMD [211]. In stage 2, our coarse-grained implementation [79] of the Weighted Histogram Analysis Method (WHAM) [234] is used to determine the probabilities of the conformations. In stage 3, cluster analysis is carried out at a temperature below the folding transition temperature, the cluster being ranked in descending order of the sums of the probabilities of the conformations constituting a cluster. The conformation closest to the cluster average is selected as the candidate prediction in the coarse-grained representation. In stage 4, the coarse-grained models obtained in stage 3 are converted to all-atom representation and refined by means of short all-atom MD with the AMBER force field. Modeling can be carried out in the free mode or with auxiliary information from templates produced by servers [235], secondary structure or contact restraints [236], as well as low-resolution experiment restraints [237]. An example of UNRES prediction of an oligomeric protein in the CASP13 exercise is shown in Figure 7.

Another example, in which the structure of the Herpes Virus Entry Mediator (HVEM) peptide complex B- and T-lymphocyte Attenuator (BTLA) was predicted in agreement with the experimental structure [238], is shown in Figure 8.

### 3.3. Investigation of Telomere Stability

Telomeres are repetitive nucleotide sequences that occur at linear DNA termini. They form unusually stable structures, in view of the fact that their sequences contain many A and T bases, which usually results in reduced stability of the double helix. These structures play a variety of roles in cell biology. Shortening telomeres keeps track of the previous cell divisions and, therefore, controls the ageing process [239]. Moreover, the loop-like or lasso-like structures of telomeres play an important role in chromosome stability [240]. Furthermore, the telomeres also adjust the cellular response to stress and growth stimulation [241].

By using steered molecular dynamics (SMD) [242] with the NARES-2P model of nucleic acids [102,147], as well as all-atom molecular dynamics with the AMBER ffSB14 force field [145,243], we attempted to explain the origin of the unusual stability of telomere structures [174]. We carried out the simulation of human-like (TTAGGG), plant (TTTAGG), insect (TTAGG) and *Candida guilermondi* (GGTGTAC) telomere repeats with various lengths, and other non-telomere repeats, AT, CG and TTTTTCCCC, for comparison [174]. The double helices were pulled either longitudinally, each strand from the other end of the helix, or pulled apart vertically, both strands at the same end of the double helix.

The force strength profiles obtained in NARES-2P simulations are shown in Figure 9A. It can be seen from the figure that all profiles exhibit a peak, with position dependent on sequence length. The profiles obtained from the all-atom simulations were largely independent of sequence, because a fast pulling force rate had to be applied, this resulting in the formation of a flat double ladder. Conversely, the faster dynamics with NARES-2P enabled us to apply a slower pulling force rate, resulting in sequence dependence of stability. As could be expected, the force peak was the highest for the CG repeats, which form the strongest Watson–Crick pairs, and the lowest for the AT repeats, which form weaker pairs (Figure 9A).

Nevertheless, it can be seen from Figure 9A that the force peaks corresponding to the telomere sequences as well as for the TTTTCCCC repeat are almost as high as that for the CG repeat. Moreover, the pulling force decays slower with stretch extent than that for the CG repeat and the profiles have a fine structure, which results from cycles of dissociation of parts of one strand from another one and subsequent “regrabbing”, as shown in Figure 9. The “regrabbing” probably is the main factor preventing chain separation and seems to be characteristic of the telomere (and the TTTTTCCCC) repeats, since it is not observed for the CG repeats. It can, therefore, be the reason for the ability of telomeres to resist damage and, thus, to protect the chromatin structure. As also found from our study, telomere resistance to forces increases significantly with repeat length. Thus, shortening the telomere upon ageing reduces its ability to protect the chromatin, finally leading to apoptosis in the cell.

## 4. Discussion

The field of coarse-grained description of the structures and dynamics of biomolecular systems is growing rapidly, both regarding the theoretical development and the scope of applications, and has advanced considerably since the seminal work by Levitt and Warshel [11]. A solid theoretical foundation of this approach has been provided by the Mori–Zwanzig theory of the projection operator [15,32,33], with which the effective equations of motion of the coarse-grained degrees of freedom could be derived [16,17,18,34] (Equation (Equation 1)). With this derivation, it is clear that the PMF of the system should be considered as the potential energy in the equations of motion, which has pave the way towards the physics-based systematic development of CG force fields, similar to the Born–Oppenheimer approximation, which enables us to express the Potential Energy Surface (PES) of a system solely in terms of the coordinates of the nuclei [6]; this has been the basis of the development of all-atom force fields. By using the factor expansion of the PMF [25,26], it is, in turn, possible to split the PMF into components that can be interpreted as specific energy terms, transferable between systems. This decomposition also enables us to derive the analytical formulas for the effective energy terms, including the multibody terms that are crucial for the correct, unassisted modeling of biomolecule structures and are not easy to obtain in other ways. The examples presented in Section 3 demonstrate that the force fields derived in such a way can model reliably the structures of proteins and the kinetics and mechanisms of protein folding, including those of post-translationally modified proteins, and the stability of DNA, including that of telomere sequences, for which the all-atom approach failed to detect the sequence dependence of stability.

There is still a long way to go until the coarse-grained force fields (and the all-atom ones as well) become so reliable as to model the structure and dynamics with experimental accuracy. The machine learning approaches [106,107,108,110,111] can be of great help in this regard, and their recent tremendous success in the knowledge-based prediction of protein structures [104,105] is a strong reason to believe that they can also lead to tremendous advancements in the development of physics-based force fields. Nevertheless, many reasonably performing coarse-grained force fields are available even at present, including the most popular MARTINI, which can be used for virtually all biologically important systems [66,67,68,69,70], Examples are UNICORN [47] and SIRAH [142,143], which also have the same scope of application, AWSEM [122] and OPEP [82,83,131] for proteins and HiRe-RNA [134] and oxDNA/oxRNA [136,137,138,139,140,141] for nucleic acids. With carefully chosen restraints, these approaches yield reliable results. Moreover, a number of force fields are available for use with MC simulations, such as, the very well performing CABS knowledge-based force field for proteins [28,31] or the SimRNA knowledge-based force fields for nucleic acids [29]. Furthermore, valuable results are also obtained with simple structure-based Gō-like models [117,118,119,120,121] or the elastic network models [114,115,116], which, although not transferable, are easy to construct for particular systems.

Another very important issue in CG dynamics is the relationship between the CG time scale and the actual time scale. Equation (Equation 1) provides the basis for this through the presence of the friction kernel and random forces. Both terms contain the whole history of the dynamics of a system and are, therefore, very difficult to handle rigorously, although research in this direction is gaining momentum [17,18]. The example presented in Section 3.1.2 demonstrates that introducing a non-diagonal solvent friction tensor (the hydrodynamic interactions) has a different effect on simulated folding depending on whether the Gō-like or physic-based CG force fields are used [42]. Furthermore, considering the non-diagonal form of the inertia tensor, which naturally arises while coarse graining biopolymers chains [21,26], as opposed to the commonly applied representation of the interaction sites as point masses [27,66], appears essential for the correct representation of the dynamics of the coarse-grained degrees of freedom. 

## Figures and Tables

**Figure 1 biomolecules-11-01347-f001:**
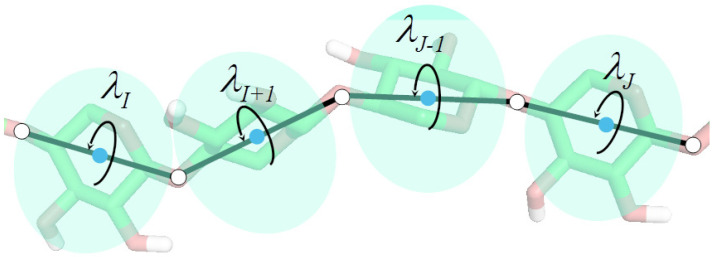
An illustration of coarse graining with the example of a chunk of poly-α-D-glucose chain. The spheroids (anchored in the glycosidic-oxygen atoms) represent united sugar rings and the angles λ for the rotation about the O⋯O virtual bond axes are the secondary degrees of freedom, which are not present in the CG model.

**Figure 2 biomolecules-11-01347-f002:**
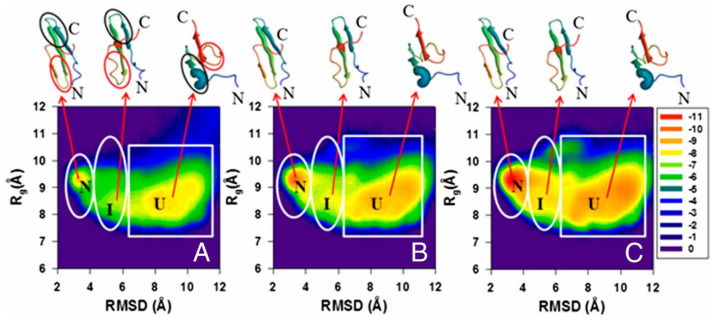
Variation of the distribution of conformational states in terms of FELs (in kcal/mol) along the C${}^{\alpha }$-RMSD and $R_{g}$ order parameters for the wild-type FBP28 WW domain. The data have been collected from different sections of all 512 trajectory sets for the molecule (shown in (**A**–**C**), respectively). The FEL corresponding to the initial parts of the trajectories (with the average fraction of the native structures up to 20% of the maximum fraction) is shown in (**A**); the FEL from the middle parts of the trajectories (the fraction of the native structures between 20% and 50% of the maximum fraction) is shown in (**B**); and the FEL from the final parts of the trajectories (the fraction of the native structures exceeds 50% of the maximum fraction) is shown in (**C**), respectively. The letters “U”, “I” and “N” correspond to unfolded, intermediate and native states, respectively. The representative structures of unfolded, intermediate and native states are plotted on top of each state. Hairpin 1 and hairpin 2 are circled by black and red lines, correspondingly, in (**A**). Reproduced from R. Zhou et al. *Proc. Natl. Acad. Sci. USA*, 111, 18243–18248 (2014). Copyright 2014 National Academy of Sciences.

**Figure 3 biomolecules-11-01347-f003:**
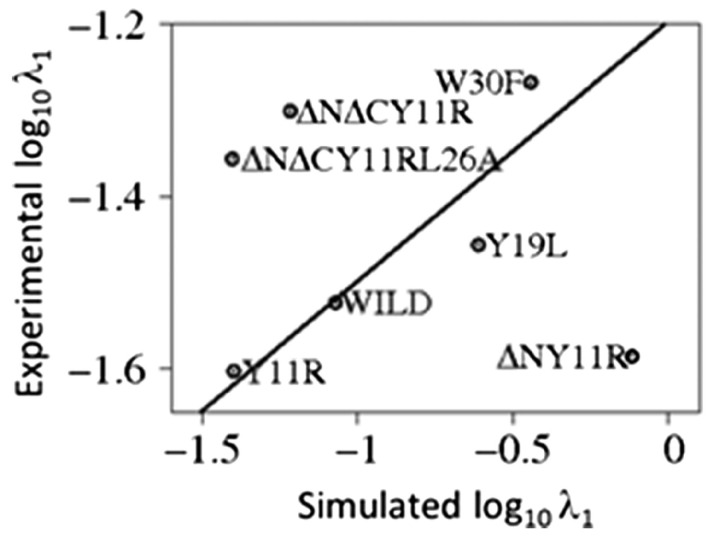
A correlation plot of the experimental and simulated fast cumulative constants with a least-squares fitting line. The experimental rate constants are expressed in μs${}^{-1}$, while those obtained from simulations are expressed in ns${}^{-1}$. Reproduced from R. Zhou et al. *Proc. Natl. Acad. Sci. USA*, 111, 18243–18248 (2014). Copyright 2014 National Academy of Sciences.

**Figure 4 biomolecules-11-01347-f004:**
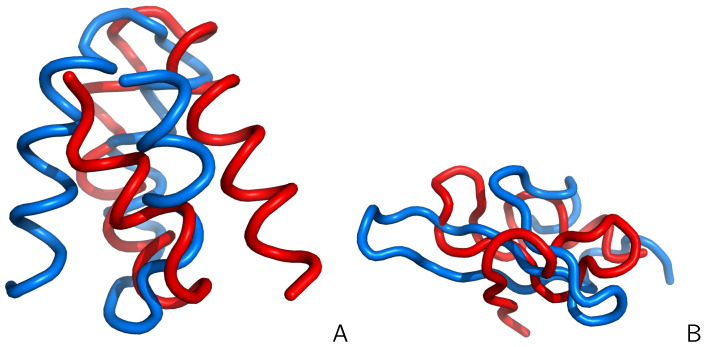
Superpositions of the experimental (blue) and intermediate (red) structures of (**A**) 1BDD and (**B**) 1E0L proteins. Reproduced from A. Lipska et al., *J. Chem. Phys.* 144, 184110 (2016), with the permission of AIP Publishing.

**Figure 5 biomolecules-11-01347-f005:**
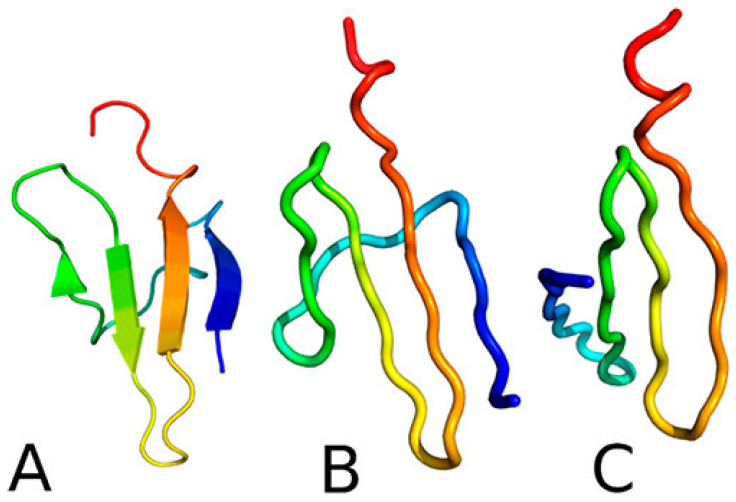
Experimental structure of pT37pT46 4E-BP2${}_{18-62}$ (**A**) compared with the most similar structures obtained after simulations and clustering of pT37pT46 4E-BP2${}_{18-62}$ (**B**) and WT 4E-BP2${}_{18-62}$ (**C**). Reproduced with permission from A.K. Sieradzan et al., *J. Chem. Theory Comput.* 17, 3203–3220 (2021). Copyright 2021 American Chemical Society.

**Figure 6 biomolecules-11-01347-f006:**
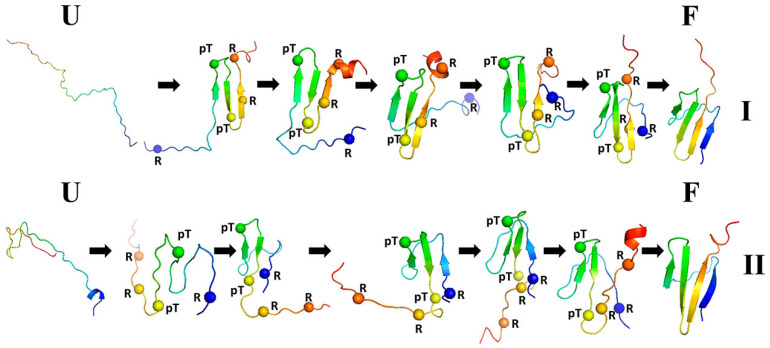
Two folding pathways of pT37pT46 4E-BP2${}_{18-62}$ found by UNRES MD simulations. Reproduced with permission from A.K. Sieradzan et al., *J. Chem. Theory Comput.* 17, 3203–3220 (2021). Copyright 2021 American Chemical Society.

**Figure 7 biomolecules-11-01347-f007:**
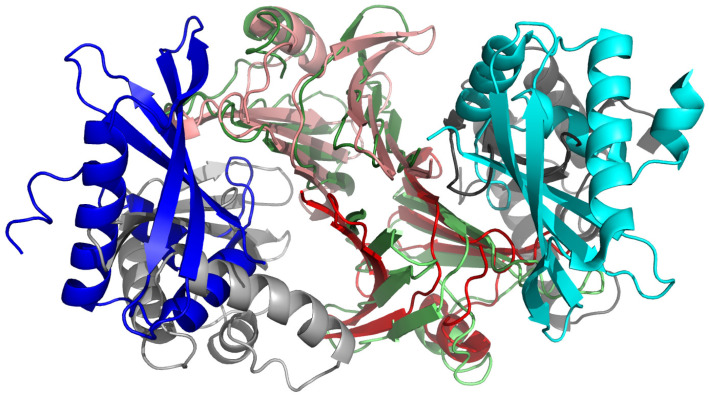
Cartoon representation the UNRES model of CASP13 oligomeric target H0968 obtained by the UNRES-based KIAS-Gdansk group (gray) superposed on the corresponding experimental structures (rainbow-colored). The interface RMSD is 1.33 Å and the model was ranked #1 among the models obtained by all groups. Reproduced with permission from A. Karczyńska et al., *J. Chem. Inf. Model.*, 60, 1844–1864 (2020). Copyright 2020 American Chemical Society.

**Figure 8 biomolecules-11-01347-f008:**
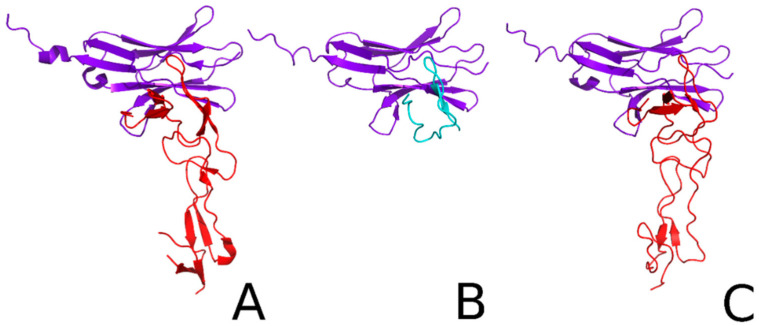
(**A**) Crystal structure of the Herpes Virus Entry Mediator (HVEM) peptide complex B- and T-lymphocyte Attenuator (BTLA) (PDB code: 2AW2), in which the purple color indicates BTLA and red indicates HVEM; (**B**) The dominant cluster of the HVEM(14–39) peptide–BTLA complex obtained from UNRES simulation, in which the purple color indicates BTLA and cyan blue indicates the peptide; (**C**) The fourth cluster of the BTLA–HVEM complex obtained from UNRES simulation, in which HVEM and BTLA are highlighted in red and purple, respectively. Reproduced with permission from M. Spodzieja et al., *Int. J. Mol. Sci.*, 21, 636 (2020) under Creative Common CC BY license.

**Figure 9 biomolecules-11-01347-f009:**
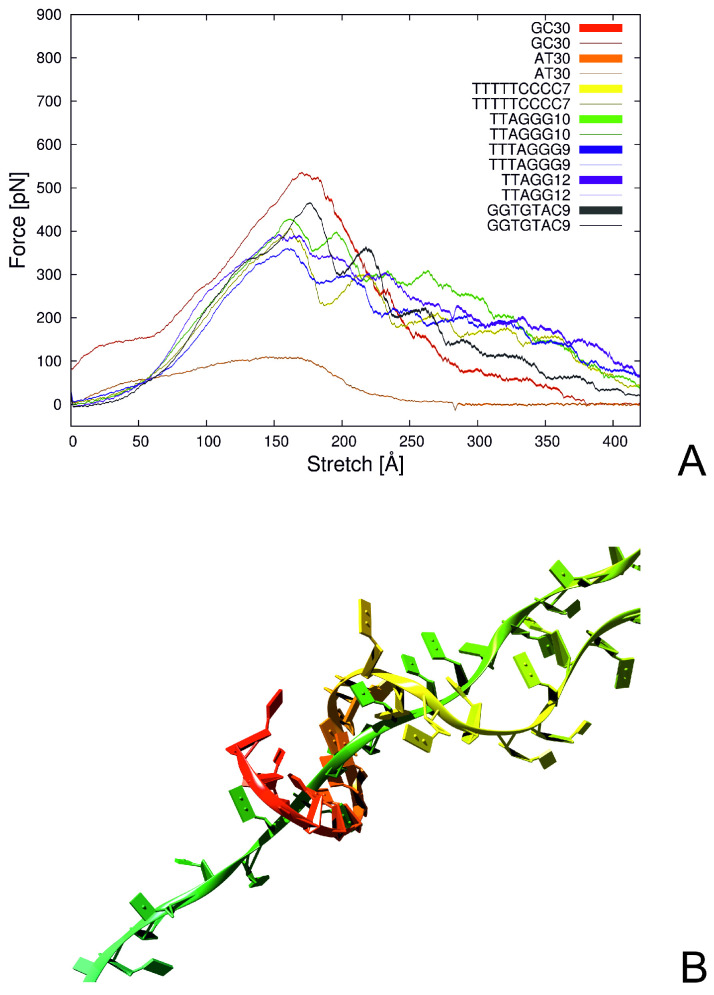
(**A**) Plot of the force needed to stretch the DNA perpendicular to the hydrogen bonds between bases, as a function of sequence; Thin lines indicate averages over 64 trajectories and the colored areas indicate one standard deviation. (**B**) Structure of telomere after regrabbing. The free end (red) wraps around the second DNA chain. Reproduced with permission from A.K. Sieradzan et al., *J. Phys. Chem. B*, 121, 2207–2219 (2017). Copyright 2015 American Chemical Society.

## Data Availability

The UNRES package is available at www.unres.pl (accessed on 10 September 2021).

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
