# Peer review of "Theory and Practice of Coarse-Grained Molecular Dynamics of Biologically Important Systems"

_biomolecules, 2021, doi:10.3390/biom11091347_

Round 1

Reviewer 1 Report

This manuscript is a review on the application of coarse-grained (CG) MD to simulate biopolymers. Given the popularity of CG MD this is a timely review that for sure will be of interest to many readers.

The paper has a clear Introduction section and a dedicated Method section outlining the main theoretical ideas behind CGing. The Results section is mostly showcasing some recent work in this area from the main authors, which is fine and serves the purpose of this review.

Overall I am happy to support this paper, but do have some comments to take into consideration to improve the overall balance and consistency, and remove or better motivate some of the biases:

1) The title is a bit misleading, as the review is primarily focussing on biopolymers (proteins, DNA), and strongly biased to solvent-free approaches in particular the UNICORN model. I would suggest to add at least the word 'biopolymer', and perhaps also 'implicit solvent'.

2) The same applies to the Introduction, which, although explicit solvent generic models as Martini are mentioned, mostly concerns solvent-free models given the emphasis on Langevin dynamics, and again, is biased towards models for protein folding. This is fine, but should then be more clearly indicated right from the beginning.

3) The Introduction section also contains the following sentence "Usually, the functional expressions are copied from all-atom force fields, which results in insufficient capacity to model the structures of the systems under study (e.g., protein structures), which puts into question the reliability of the modeled dynamics, ensemble averages, and properties in general." which seems a bit harsh on CG approaches that (successfully) rely on these functional forms.

4) I find the naming of sections 2 (Materials and Methods) and 4 (Results) a bit misleading for a Review article.

5) Line 220: Berendsen is misspelled. In the context of stochastic thermostats,  another recent and widely used thermostat developed by Berendsen and coworkers could be cited: Goga et al., JCTC 8, 3637, 2012

6) Lines 270-272 state: "Moreover,  the site-site interaction potentials of most coarse-grained force fields are usually too sticky, this resulting in too compact modeled structures." This is a weird statement, and probably originates from the stickiness reported for the Martini model. However, for this model, the reasons for the stickiness are different, as recently discussed in a paper by Alessandri et al. (JCTC 15, 5448, 2019). Moreover, these problems were overcome with the most recent reparameterization effort of the model (Souza et al, Nature Methods, 18, 382, 2021). These papers should be cited to provide a fair view on the Martini model.

7) Section 2.3.2: apart from the citations mentioned above, it is noted that Martini has also been extended into materials science (Alessandri et al., Advanced Materials 33, 2008635, 2021). The statement 'four to eight' water molecules is wrong, standard Martini only uses a mapping of four waters into one CG bead. Furthermore, it is unclear what is meant with 'the force field is not predictive' - presumably the authors refer here to 2ndary structure prediction, but this needs to be specified.

8) Section 2.3.7 is very odd and does not seem to fit very well with the rest of the review. Why a special section about this particular sub-class of biomolecules ?

9) Section 3 is mostly about REMD and related exchange methods. Why are other enhanced sampling techniques not covered here ? With respect to exchange methods, one could also mention replica exchange schemes using CG models to speed up all-atom dynamics, see e.g. recent work from Liu et al., JCTC 16, 5313, 2020).

Author Response

We thank the Reviewer for his/her careful evaluation of our manuscript and very meaningful suggestions. We revised the manuscript accordingly. Our responses and description of revisions are detailed below, with page and line numbers of the revised manuscript in which revisions have been performed listed. For clarity, we quote the comments before each response. We also provide a version of the manuscript with changed text shown in red font.

1) The title is a bit misleading, as the review is primarily focussing on biopolymers (proteins, DNA), and strongly biased to solvent-free approaches in particular the UNICORN model. I would suggest to add at least the word 'biopolymer', and perhaps also 'implicit solvent'.

We changed the title to „Theory and practice of coarse-grained molecular dynamics of biologically-important systems” to indicate the class of problems covered. However, our article discusses both implicit- and explicit-solvent approaches, as explained in the next resposes and, therefore, we did not include the phrase „implicit solvent” in the revised title.

2) The same applies to the Introduction, which, although explicit solvent generic models as Martini are mentioned, mostly concerns solvent-free models given the emphasis on Langevin dynamics, and again, is biased towards models for protein folding. This is fine, but should then be more clearly indicated right from the beginning.

In the Introduction we added a phrase stating that the emphasis is given to biolopolymer systems (page 2 line 69). As stated in section Langevin dynamics is an approximate treatement of the friction and stochastic forces [equations (4) and (5)]. Therefore, Langevin dynamics should be used also when the solvent is explicit. The friction and stochastic forces always appear in CG equations of motion [equation (1)] as a result of neglecting the fine-grained degrees of freedom, even if explicit solvent is used. One part of this contribution comes from internal friction, which appears even without any solvent. However, another part comes from the rotational degrees of freedom of the solvent molecules, which are averaged over in the explicit-solvent CG MD. Moreover, Langevin dynamics has also been recommended for use with all-atom MD with explicit solvent, because it provides better thermostating compared to Berendsen (Paterlini and Fergusson, Chem. Phys., 236, 243-252, 1998). The above considerations have been included in the revised manuscript (page 4 lines 137-148; page 6, lines 216-219).

3) The Introduction section also contains the following sentence "Usually, the functional expressions are copied from all-atom force fields, which results in insufficient capacity to model the structures of the systems under study (e.g., protein structures), which puts into question the reliability of the modeled dynamics, ensemble averages, and properties in general." which seems a bit harsh on CG approaches that (successfully) rely on these functional forms.

We softened the quoted sentence (page 2, lines 64-66) to state that a force field can be useful even if it cannot model structures, if applied with caution and if the results are critically analyzed.

4) I find the naming of sections 2 (Materials and Methods) and 4 (Results) a bit misleading for a Review article.

We changed the titles of the „Materials and Methods” section to to „Theory and Methodology” and that of the „Results” section to „Examples”.

5) Line 220: Berendsen is misspelled. In the context of stochastic thermostats,  another recent and widely used thermostat developed by Berendsen and coworkers could be cited: Goga et al., JCTC 8, 3637, 2012

The spelling error has been corrected and the suggested reference added (new reference 54).

6) Lines 270-272 state: "Moreover,  the site-site interaction potentials of most coarse-grained force fields are usually too sticky, this resulting in too compact modeled structures." This is a weird statement, and probably originates from the stickiness reported for the Martini model. However, for this model, the reasons for the stickiness are different, as recently discussed in a paper by Alessandri et al. (JCTC 15, 5448, 2019). Moreover, these problems were overcome with the most recent reparameterization effort of the model (Souza et al, Nature Methods, 18, 382, 2021). These papers should be cited to provide a fair view on the Martini model.

We revised the sentence and the adjacent text fragments to state that the „stickiness” problem in Martini has been overcome, cited the suggested references (page 7, lines 284-292). The suggested references have been cited (new references 76 and 77).

7) Section 2.3.2: apart from the citations mentioned above, it is noted that Martini has also been extended into materials science (Alessandri et al., Advanced Materials 33, 2008635, 2021). The statement 'four to eight' water molecules is wrong, standard Martini only uses a mapping of four waters into one CG bead. Furthermore, it is unclear what is meant with 'the force field is not predictive' - presumably the authors refer here to 2ndary structure prediction, but this needs to be specified.

We corrected the fragment specifying the number of water molecules in the Martini water site (page 10, linex 402 and 403) and added a statement that Martini has been extended to materials science and cited the suggested reference (page 9, line 396). We also clarified the sentence about the structure-prediction ability of Martini (page 10, lines 408-409).

8) Section 2.3.7 is very odd and does not seem to fit very well with the rest of the review. Why a special section about this particular sub-class of biomolecules ?

Glycosoaminoglycans are a very important constituent of extracellular matrix but seem to be largely ignored in reviews on simulation methodology and applications, probably because more extensive simulation studies on these molecules have been started quite recently. Therefore, we would like to keep this section. We revised the pertinent fragments of the text to state why the discussion of the CG force fields that treat these molecules, in our opinion, worthwhile (page 9, lines 365-367; page 11, lines 501-502).

9) Section 3 is mostly about REMD and related exchange methods. Why are other enhanced sampling techniques not covered here ? With respect to exchange methods, one could also mention replica exchange schemes using CG models to speed up all-atom dynamics, see e.g. recent work from Liu et al., JCTC 16, 5313, 2020).

REMD and MREMD are probably the most popular MD extensions that are used with CG models and, therefore, we put emphasis on these methods. However, we also briefly discuss multicanonical algorithms. In the revised manuscript, we cited the suggested reference (new reference 216) and discussed its content (page 13, lines 599 and page 14, lines 600-601). We also mentioned more MD extensions used with CG MD (page 12, lines 552-556; page 13, lines 557-571; page 14, lines 607-610). Also, we made the former section 3 a part of the Theory and Methodology subsection (i.e., subsection 2.4 of the current section 2).

Reviewer 2 Report

The authors did a good job in reviewing the general methodology of coarse-grained molecular dynamics. They also reviewed a few examples of popular coarse-grained models for biomolecule systems followed by the discussion on specific examples such as folded protein and DNA. There are a few questions listed below for the author to address.

    The authors mention the event-driven discontinuous molecular dynamics first introduced by Alder and Wainwright in the introduction section. The PRIME20 protein model (DOI: 10.1002/prot.22817 and 10.1002/prot.1100) is a knowledge-based 4-bead-per-residue CG model using square well and hard sphere potentials tailored to model large-scale protein misfolding and aggregation problems, which may also be worth to mention and discussed in the text.

    On page 12, the author have some discussions on enhanced sampling techniques that help improve the sampling efficiency of canonical or isobaric MD simulation. In addition to the replica-exchange method, some other methods such as metadynamics and forward flux sampling, may also be mentioned and briefly discussed.

Minor comments:

  1. On page 1, line 32, "Alder" is misspelled as “Adler”.
  2. The two figures in Fig. 9 are not aligned as well as the corresponding A and B labels, and may be corrected.

Author Response

We thank the Reviewer for his/her careful evaluation of our manuscript and very meaningful suggestions. We revised the manuscript accordingly. Our responses and description of revisions are detailed below, with page and line numbers of the revised manuscript in which revisions have been performed listed. For clarity, we quote the comments before each response. We also provide a version of the manuscript with changed text shown in red font.

„    The authors mention the event-driven discontinuous molecular dynamics first introduced by Alder and Wainwright in the introduction section. The PRIME20 protein model (DOI: 10.1002/prot.22817 and 10.1002/prot.1100) is a knowledge-based 4-bead-per-residue CG model using square well and hard sphere potentials tailored to model large-scale protein misfolding and aggregation problems, which may also be worth to mention and discussed in the text.

We have cited the suggested references in the revised manuscripts and added pertinent discussion (page 6, lines 232-235, new references 57 and 58).

„    On page 12, the author have some discussions on enhanced sampling techniques that help improve the sampling efficiency of canonical or isobaric MD simulation. In addition to the replica-exchange method, some other methods such as metadynamics and forward flux sampling, may also be mentioned and briefly discussed.

The pertinent discussion has been added (page 12, lines 552-556; page 13, lines 557-571 and 599; page 14, lines 600-601 and 607-610) and the appropriate references cited. Also, we made the former section 3 a part of the Theory and Methodology subsection (i.e., subsection 2.4 of the current section 2).

Minor comments:

    1.  

On page 1, line 32, "Alder" is misspelled as “Adler”.

    1.  

The two figures in Fig. 9 are not aligned as well as the corresponding A and B labels, and may be corrected.”

These issue have been fixed.

Reviewer 3 Report

This manuscript is a detailed review paper on the uses and development of coarse-grained simulation. It starts from very fundamental basics on coarse-grained simulations, explains the pros and cons of available force-fields, and introduced simulation results for the investigation of protein folding and stability of DNA. This review paper could be useful to readers who are interested in coarse-grained biomolecular simulation. A weak point of this manuscript is that the review seems focused on their own works.

Author Response

We thank the Reviewer for his/her careful evaluation of our manuscript and very meaningful suggestions. We revised the manuscript accordingly. Our responses and description of revisions are detailed below, with page and line numbers of the revised manuscript in which revisions have been performed listed. For clarity, we quote the comments before each response. We also provide a version of the manuscript with changed text shown in red font.

This manuscript is a detailed review paper on the uses and development of coarse-grained simulation. It starts from very fundamental basics on coarse-grained simulations, explains the pros and cons of available force-fields, and introduced simulation results for the investigation of protein folding and stability of DNA. This review paper could be useful to readers who are interested in coarse-grained biomolecular simulation. A weak point of this manuscript is that the review seems focused on their own works.

We added some more discussion of other methods for conformational sampling, which are not used in our laboratory (page 12, lines 552-556; page 13, lines 557-571 and 599; page 14, lines 600-601 and 607-610 of subsection 2.4 of section 2). Also, in section 2, apart from our work, we discuss the commonly used approaches to CG MD, in particular the commonly applied CG force fields (subsections 2.3.1 – 2.3.7).
